# "Jewish Mindfulness" as Spiritual Didactics Teaching Orthodox Jewish Religion through Mindfulness Meditation

**Mira Niculescu** 

César, EHESS, 75006 Paris, France; nicu@ehess.fr; Tel.: +336-0176-5607

**Abstract:** Since the late 1990s, the expression "Jewish Mindfulness" has become ubiquitous in Jewish community centers (JCCs) and synagogues in America, in Israel, and in the Western diaspora. "Mindfulness", a secular meditation technique originating from Buddhism which has been popularized in Western culture through its recontextualization within the Western therapeutic culture, has been increasingly used in Jewish Religious settings, including Modern Orthodox. How do Modern Orthodox rabbis describe their use of "Mindfulness" in their religious teachings? Why do they refer to Mindfulness Meditation rather than to Jewish Meditation? In this article, I comparatively analyze the discourses spoken—online, and in print—of American rabbis from various Modern Orthodox trends as a case to study strategies of adaptation in the current context of globalization. By identifying three types of use of Mindfulness—*through, and* or *as* Judaism—I seek to highlight the various ways in which today's Orthodox educators use "Mindfulness", both as a meditation technique and as a spiritual mindset, and how this is reshaping the way they teach Jewish religion. Observing contemporary Orthodox discourses on Mindfulness within Jewish religious pedagogy can help us better understand the processes of cultural appropriation and translation as well as religious change in the making, as part of a boundary maintenance work within today's cosmopolitan cultures.

**Keywords:** Judaism; mindfulness; meditation; Buddhism; religion; spirituality; symbolic boundaries; pedagogy; cosmopolitanism; globalization

---

*"The Jews have not merely a tendency to imitation, but a genius for it."* (Ahad Ha'am 1912)

## 1. Introduction: Understanding Cultural Appropriation through the Case of Modern Orthodox Uses of Mindfulness Meditation in Jewish Education

One may be surprised to find that upon entering the Heschel Jewish High School[1] in New York at the time of morning prayers, there is an option to participate in a "Mindfulness and meditation minyan[2]". Mindfulness, although known today in Western culture as a secular meditation technique consisting simply in "paying attention" to "the present moment[3], "is nevertheless derived from *Vipassana*[4], a Modernized form of South-East Asian Theravada Buddhism, which takes ground in a foundational Buddhist text, the *Satipaṭṭhāna Sutta*[5]. Could these Jewish educators be suggesting the

---

1. The Heschel School: https://www.heschel.org/ (accessed on 1 December 2019).
2. The "minyan" is the quorum of ten men traditionally required for a prayer service. Today, the term has become generic for a prayer group.
3. https://www.mindful.org/jon-kabat-zinn-defining-Mindfulness/ (accessed on 27 April 2019).
4. Jay Michaelson "The Man who taught the World to meditate" *Huffington Post*, June 30, 2013. Available online: https://www.huffpost.com/entry/sn-goenka-dead_b_4016374 (accessed on 13 December 2019).
5. In the terms of the Pāli language it was first written into.

practice of a Buddhist form of meditation instead of Jewish prayer? Not at all, and, they would argue, on the contrary. First of all, in Western culture, "Mindfulness" is now unambiguously considered to be a secular meditation technique, thereby not competing with any religious practice. Second of all, and most importantly so, the goal of such an alternative minyan is not to substitute "meditation" to "prayer," or "Mindfulness" to "Judaism." Quite the opposite, says the description on the school's website; it is to "develop Mindfulness skills in the context of Jewish tradition and Tefilah[6]." In other words, the intent is to use Mindfulness as a pedagogic tool to better access Jewish prayer (*tefilah*).

### 1.1. The Mindfulness Revolution and the Orthodox Ethos of Boundaries

Not all Jewish day schools in America have added Mindfulness to their prayer curriculum. Yet more and more of them seem to have followed Heschel's lead and are now offering various forms of "Mindfulness minyan" options[7]. In fact, the trend seems to be have gained so much momentum that various forms of trainings in Mindfulness for Jewish Educators are now being offered. Following the pioneering project offered by the Institute for Jewish Spirituality (IJS) entitled "Educating for a Spiritual life[8]," Jewish organizations such as the "Jewish Education Project[9]" as well as private consultants[10], are now offering Mindfulness workshops and trainings for Jewish Educators.

Replaced in the current cultural context of American society, this phenomenon should not come as a surprise: it is part of a wider cultural trend, by which "Mindfulness" has become highly popular in Western culture at large, and in America in particular (Wilson 2014)—so much so that in 2014, the magazine *Time* dedicated its cover to "The mindful revolution[11]". The success of "Mindfulness" in Western culture is due to the fact that it is presented simply as "the science of finding focus in a stressed-out, multitasking society[12]." Decontextualized from its original Vipassana context and recontextualized within a Western Scientific perspective, "Mindfulness" has been construed as a secular therapeutic meditation technique on its own. As a consequence, it has become ubiquitous in contemporary Western society: it is nowadays being taught in hospitals and in prisons (Fronsdal 1998), in political institutions (Ryan 2012) and in Silicon Valley corporations (Tang 2012), but also, perhaps more surprisingly, in religious contexts. The phenomenon of Jewish Mindfulness is but one aspect of a broader phenomenon of the appropriation of Mindfulness within Western religious contexts. One can observe, for example, similar trends in the phenomenon of "Christian Mindfulness" (Tan 2011; Symington and Symington 2012; Frederick and White 2015; Trammel 2015; Ford and Garzon 2017) and "Muslim Mindfulness" (Ghorbani 2009; Mirdal 2012; Thomas et al. 2016; Helminski 2017; Salyers 2017).

Within the Religious Jewish field, beyond the case of day schools, over the past half-decade, several indicators tend to show that this emerging trend is growing: first, the spread of the practice in mainstream Jewish settings : Mindfulness programs in local Jewish community centers (JCC's[13]),

---

[6]    "Tefilah" (option n°5) *Heschel*. Available online: https://www.heschel.org/academics/high-school/tefillah (accessed on 1 December 2019).

[7]    See: Dan Finkel, 2018, "Jewish Mindfulness" *Gesher Jewish day school*. Available online: https://www.gesher-jds.org/2018/11/09/Jewish-Mindfulness/ (accessed on 27 April 2019); "Mindfulness and Jewish Spirituality". *Gesher Community Day School*. Available online: https://www.gesher-jds.org/2018/11/09/Jewish-Mindfulness/ (accessed on 27 April 2019); "Mindfulness as School Practice: A Conversation with Oakland Hebrew Day School's Tania Schweig". *Prizmah*. Available online: https://medium.com/@Prizmah/Mindfulness-as-school-practice-a-conversation-with-oakland-hebrew-day-schools-tania-schweig-77e0ede4452b (accessed on 27 April 2019).

[8]    "Educating for a Spiritual life" *Institute for Jewish Spirituality*. Available online: https://www.Jewishspirituality.org/teach-others/educating-for-a-Jewish-spiritual-life/ (accessed on 1 December 2019).

[9]    "Mindfulness for Day School Educators," *The Jewish Education Project*. Available online: https://www.Jewishedproject.org/events/Mindfulness-day-school-educators (accessed on 1 December 2019).

[10]   Nancy Siegel Consulting "Mindfulness for Educators". Available online: https://nancysiegelconsulting.com/programs-for-educators (accessed on 1 December 2019).

[11]   Kate Pickert, "The mindful revolution." *Time*, February 3, 2014. Available online: http://content.time.com/time/covers/0,16641,20140203,00.html (accessed on 27 April 2019).

[12]   As the cover subtitle indicates.

[13]   "Makom" Meditation and Mindfulness, Manhattan JCC: https://jccmanhattan.org/makom; Makor Or, San Francisco JCC: https://Jewishfed.org/news/events/makor-or-center-Jewish-meditation-high-holidays-preparation-retreat.

but also in local Synagogues and Yeshivot and the organization of Jewish Mindfulness retreats and training at global Jewish youth organizations (Hillel[14], Moishe House[15]) have been multiplying over the past decade; second, the institutionalization of the practice: local[16] and global Not-for-Profit Organization dedicated to Jewish Mindfulness and Contemplative practice[17] have emerged and grown in the same time period; third, the professionalization of the practice: Jewish Mindfulness Teacher Training programs[18] are now being offered to Jewish Educators; and fourth, the diffusion of resources and the publicization of the practice, with the exponential publication of books (Slater 2004; Michaelson 2007; Shaw 2017; Epstein 2019) and articles ,in the Jewish Press and on various Websites,[19] about "Jewish Mindfulness", and the broadcasting of videos on YouTube, as the number of Web entries dedicated to this topic show[20].

Yet if it has developed spectacularly over the past decade, the phenomenon did not start overnight. It is rather the continuation of the progressive import of Mindfulness, along with other Eastern-derived spiritual practices, within the Jewish religious field over the last half-century. Since Rabbi Zalman Schachter Shalomi, the founder of the Jewish Renewal Movement, integrated Vipassana meditation to his *Yom Kippur* services in 1973, a long journey has occurred. Today, Mindfulness is not only part of renewal or liberal settings, it has entered Orthodox settings, especially the Modern Orthodox world[21]. While Heschel, which pioneered the concept five years ago, is a non-denominational school, just a few miles away, Salanter Akiba Academy (SAR[22]), one of the largest Modern Orthodox Jewish Day Schools of the East Coast, has likewise started to implement meditation minyanim to make Jewish prayer more accessible to its students[23].

Why do Jewish educators use Mindfulness to teach Jewish religion? What makes Orthodox teachers, who, from their own educational background, certainly do not ignore the existence of Jewish meditation and who would have the skills to teach it, rather focus on Mindfulness[24]?

---

14  Hillel's "Jewish Mindfulness Fellowship": https://www.pennhillel.org/events/Jewish-Mindfulness-fellowship (accessed on 27 April 2019).

15  Moishe House "Jewish Mindfulness retreats": https://www.moishehouse.org/find-a-retreat/Jewish-Mindfulness/; https://www.moishehouse.org/find-a-retreat/Jewish-Mindfulness-intensive/ (accessed on 27 April 2019).

16  For instance: in Chicago, *Mishkan*: https://www.mishkanchicago.org/event/om-Jewish-Mindfulness-collective-2/, in London, Alyth: https://www.alyth.org.uk/be-with-us/synagogue-groups/Jewish-Mindfulness-meditation/, in England, Hamakom: http://www.hamakom.org.uk/.

17  In the United States and in Israel: in New York, the Awakened Heart Project: https://www.awakenedheartproject.org/, in Israel and the United States: *Or ha lev*: https://www.orhalev.org/.

18  "Jewish Mindfulness teacher training": https://www.Jewishspirituality.org/jmtt3/.

19  Jay Michaelson, February 3, 2014 "Why Jews should tune in to the Mindfulness revolution" *The Forward*, http://forward.com/articles/192109/why-Jews-should-tune-in-to-the-Mindfulness-revolut/#ixzz2sUWS8SLl (accessed on 27 April 2019); Michelle Goldberg, "The roots of Mindfulness *Tablet*, available at: https://www.tabletmag.com/Jewish-life-and-religion/193989/the-roots-of-Mindfulness (accessed on 27 April 2019); Simon Rocker, "Finding a Jewish path to Mindfulness", *The Jewish Chronicle*, May 12 2019, available at: https://www.thejc.com/judaism/features/finding-a-Jewish-path-to-Mindfulness-1.484017 (accessed on 27 April 2019).

20  A bracketed "Mindfulness in Jewish day schools" entry on Google receives 8 340 000 responses: https://www.google.com/search?sxsrf=ACYBGNQh1b1vHXgFT_iwy-h8FW2HjCUOlw%3A1575351444075&ei=lPTlXbKeBPLTgwfmxazoDg&q=%22Mindfulness+in+Jewish+day+schools%22&oq=%22Mindfulness+in+Jewish+day+schools%22&gs_l=psy-ab.3..33i160l2.13599.14568..14891...0.0..0.138.465.0j4......0....1..gws-wiz.......35i39.S869MaDwr6Y&ved=0ahUKEwiyuOG54ZjmAhXy6eAKHeYiC-0Q4dUDCAs&uact=5 (Accessed on 1 December 2019).

21  I leave aside other type of Orthodox groups such as Chabad, who have also heavily invested the Topic of Jewish Mindfulness (See for instance Laibl Wolf "Jewish Mindfulness. A practical Session" Chabad. Available online: https://www.chabad.org/multimedia/media_cdo/aid/1892097/Jewish/Jewish-Mindfulness.htm (accessed on 27 April 2019). Such teachers who do not define as "Modern Orthodox", use what is popular in their surrounding culture (the terminology of Mindfulness) as part of their reach-out effort (see Niculescu 2014). I want to focus on Orthodox teachers who actually use Mindfulness meditation as such and for themselves before teaching it.

22  Salanter Akiba Academy: https://www.saracademy.org/ (accessed on 1 December 2019).

23  Ben Harris "Minyans for meditation, artists and doubters: How Jewish day schools are reimagining daily prayer" the Jewish Telegraphic Agency (JTA), September 25, 2019: https://www.jta.org/2019/09/25/united-states/minyans-for-meditation-artists-and-doubters-how-Jewish-day-schools-are-reimagining-daily-prayer (accessed on 1 December 2019).

24  Most of the time in practice, they also use traditional Jewish meditation techniques. But "Mindfulness" remains constant and central both in terms of labeling and content of this new Jewish spiritual pedagogy.

The recent and generalized use of "Mindfulness" in the world of Orthodox Jewish Religious Teaching poses once again questions on cultural appropriation and its boundaries.

Such questions are not trivial from a Jewish perspective, and even more so from an Orthodox Jewish perspective. Judaism, as a social system, is based on the notion of (hermetic) symbolic boundaries (Lamont and Fournier 1992), aimed at preventing any external influence, which represents a danger for the integrity of the system itself (Douglas 1966). Yet cultural borrowing is also part of the way Jewish Culture has built itself. As a consequence, throughout centuries of diaspora, rabbinic Judaism has maintained a delicate balance for itself in the interplay between claims of boundary maintenance on the one hand, and processes of cultural appropriation on the other, (Boyarin 2004; Berkowitz 2012; Haskel 2016). Likewise, Orthodox Judaism, a Modern movement which emerged in the late Eighteenth century as a response to the post-Enlightenment secularization of European Jews (Katz 1986; Singer 1989; Silber 1992; Ferziger 2005; Ravitsky 2007), while it defines itself as faithful, religiously, to Halakha (Jewish law), and culturally to the Jewish "tradition[25]", unfolds in a constant creative tension between various levels of protectionism and various levels of opening for the sake of acculturation. It is less surprising that the emerging actors of this trend would be Modern Orthodox Rabbis rather than more stringent Orthodox rabbis: the Modern Orthodox, as Heilman and Cohen (1989) have aptly put it, are both "Cosmopolitans and Parochials"—they strive to embrace both their surrounding culture and the Jewish tradition. And Mindfulness is now fully part of their surrounding culture.

Hence, the current phenomenon of the import of Mindfulness within the world of Orthodox religious Jewish teaching is but the latest outcome of an old pattern, within the contemporary context of the encounter of Western Jews with "global Buddhism" (Baumann 2001). The center stage of this encounter is in America, where Jews have been particularly active in the acculturation of Buddhism within Western culture (Layman 1975, p. 257; Linzer 1996; Fronsdal 1998; Prebish and Tanaka 1998; Seager 1999; Coleman 1999; Niculescu 2015a, 2017b; Sigalow 2019).

But if the appropriation of Buddhism, by the youth of the "Generation of seekers" (Roof 1993) of the Counter-Culture, giving birth to the phenomenon of the "Jewish Buddhists", could be described as a bottom–up phenomenon, the crafting of Jewish Mindfulness appears as a top–down one: it is formulated by Rabbis, who write and teach about it as Jewish educators and theologians.

While the Jewish History of cultural contacts and cross-fertilization has been studied in the case of the Encounter with Christianity (Liebes 1983; Boyarin 2004; Magid 2014) and Islam (Idel 1988, pp. 73–89; Reiser 2018), the question of the more recent encounter with Buddhism is just beginning to be addressed. Within the Scientific literature focusing on Jewish education, while books such as the *International Handbook of Jewish Education* briefly address topics such as "spirituality" (Shire 2011) or the "encounter of Judaism with other religions" (Gillis 2011), there is neither mention of "meditation" nor of "mindfulness." As for the sociological and anthropological literature addressing the question of the appropriation of Mindfulness within Jewish religious settings, scholars have focused up until now on non-Orthodox settings, either and mostly on the Jewish Renewal or New Age scene, respectively in North America (Ariel 2003; Rothenberg and Vallely 2008; Weissler 2011) and in Israel (Ruah-Midbar 2012; Werczberger 2017), or on various forms of Conservative, Liberal, Reconstructionist and Non-Denominational Movements, in the United States (Sigalow 2019) as well as in a global perspective (Niculescu 2013, 2015a, 2015b).

In this article, I would like to take the direction of this research in two yet-to-be explored ways: first, I focus on the Modern Orthodox appropriations of Mindfulness; and second, I tackle the analysis from a "glocal" (Robertson 1992, p. 173) perspective, rather than just a local one. I will be seeking to understand how Jewish educators who are Modern Orthodox, in America, in England and in Israel, are justifying the integration of Mindfulness within their Jewish practice. And I will be highlighting how they claim to do so, not at the expense of their faithfulness to Jewish tradition, but precisely

---

[25]　Being thereby, according to Historian Katz (1986), "traditionalists" rather than "tradition-bound.".

*because* of it: how they practice and teach mindfulness *in order to revive* Jewish spirituality within Jewish practice, and to promote it in their educational capacity.

*1.2. Five Modern Orthodox Agents of Change and Three Strategies of Jewish Appropriation of Mindfulness*

From a perspective of anthropology of globalization, I consider, with Hahn (2008, p. 197), that it is more relevant to speak about cultural "appropriation" rather than "diffusion"—especially in the case of the study of the integration of Mindfulness within contemporary Western Jewish religious culture. Indeed, while other scholars have talked about this phenomenon as a case of "cultural diffusion" (Sigalow 2019), I argue that the phenomenon we are currently observing is one of cultural appropriation through cultural translation. Indeed, I suggest that the current integration of Buddhism and of Buddhist-derived concepts and practices in the West is no more the result of the Missionary work of Buddhist teachers—as it was in the first half of the Twentieth century—but is rather the result of strategic appropriation from various actors in Western society. In order to better grasp and understand this process of cultural appropriation in its bi dimensionality, while I zoom in to focus on the subjective perspective of Modern Orthodox Rabbis, I keep in sight the global dimension of the phenomenon. This is the "glocal" approach framed descriptively by Robertson (1992) and advocated methodologically by Hahn (2008). Indeed, as we will see, most of the actors[26] selected for this research are part of the same global network of Jewish Mindfulness (Niculescu 2015b)—of which ,they are active actors, if not leaders. As such, most of them know each other and at least of each other.

Such a "micro-macro" perspective allows both for understanding individual/local strategies, and for identifying the broader global trends that influence them. My PhD dissertation on the phenomenon of the Jewish Buddhists in a global perspective has served as a preliminary research for this paper. Since 2008, I have been observing, through participant observation in Jewish communities in the United States, in Israel, in France and in England, the unfolding of this global phenomenon, and the emergence of new actors and leaders in the field of "Jewish mindfulness". Some of the actors interviewed here I knew from having followed their trajectory as part of my anthropological study about spiritual trajectories. Others I have met through participative observation, or from reading their discourses published online or in print. Indeed, for this paper, I have searched for publications bringing together the words "Judaism" and "Mindfulness." Considering, with Barth (1969), that in order to observe cultural change in social groups, it is most heuristic to focus on "agents of change:" structurally influential individuals who, through their subjective choices and through the public positioning thereof, have an impact on the group at large[27], I have chosen to focus on displaying and analyzing the discourses of a selected number of individuals (six) who, from their positioning on the Jewish field and their discourses, represent "Agents of change" in the field of Jewish mindfulness. I have tried to gather a sample that is representative of what this field is like today: a field still widely lead by American rabbis and expanding from there in Israel and in the English-Speaking diaspora.

As such, of the six Modern Orthodox Rabbis selected as a sample for this study, three of them are American (Ayal Robkin, Hillel Brody, Ariel Evan Mayse), two American-Israeli (James Maisels and Sam Feinsmith), and one British (Samuel Landau); four of them currently live in various parts of America (Robkin and Brody in New York, Feinsmith in Chicago, Mayse in Stanford) and two of them in Israel (Maisels and Landau). They come from similar socio-cultural and religious backgrounds: born between the seventies and the nineties to Ashkenazi families, they all come from middle–upper-class educated families, they all have University degrees (and for three of them, a Doctorate), as well as, except for Brody, a Rabbinic Ordination), they are almost all married with children. On the level of their religious and spiritual life, they have all received more or less formally, and more or less consistently, a Modern Orthodox education, and have all studied in Yeshiva in Israel or in the United states where they received

---

26　I speak, with Hahn, in terms of "actors", in order to stress the agency of local actors.
27　I have spoken of "Jewish Buddhists" as agents of change in a previous article (Niculescu 2017a).

Modern Orthodox ordination from various trends of Orthodox rabbinic authorities. They have all been exposed to Mindfulness meditation during their College years, either, for the ones born closer to the seventies, within Buddhist contexts[28], or, for those born closer to the nineties, within Jewish contexts. They have all been in a spiritual search, and for most of them, dissatisfied at some point with Judaism but without ever leaving formally Jewish religious settings. They all feel that bringing Mindfulness into their Jewish Practice has brought them closer to Jewish Spirituality, and they all seem to have found a home, whether they say it explicitly or not, in what they all call "neo-Hasisidm". Whether they call themselves today explicitly "Modern Orthodox" or not, they all have rabbinic ordination from Modern Orthodox Rabbis, and they all claim to follow Halakha (Jewish law). They are all having a creative impact on the world of Jewish education today, in terms of bringing mindfulness practice within traditional Jewish learning, either through creating or implementing educational programs (Feinsmith, Robkin), or through having started Jewish mindfulness organizations (Maisels, Feinsmith), or through having published articles or books theorizing their use of mindfulness to experience and teach Jewish religion (Landau, Maisels, Mayse). Finally, as stated before, they almost all know each other from having learned from or been colleagues to each other. For these reasons, from their structural position as well as from their action within the world of Jewish education, they all play the part of "agents of change."

In this article, I focus on the ways a small number of Modern Orthodox rabbis, because of their innovative approach and their strategic positions in the field of Jewish Education, are playing the role of "agents of change" triggering innovation in the way Jewish spirituality is being taught all the while reaffirming their faithfulness to the symbolic boundaries of the Jewish tradition. I describe the ways they do so by appropriating, translating and integrating mindfulness within their teachings of Jewish spirituality within an Orthodox frame in creative ways. I use the case of the integration of mindfulness within the field of Orthodox religious teaching as a case to study boundary-maintenance through boundary shifting, and tradition maintenance through creativity.

In each chapter and sub-chapter of this article, I try to weave their discourse together, following a thematic typology which has emerged from the content analysis of their discourses. Indeed, following the inductive approach characteristic of qualitative research, I have analyzed their discourses through word and thematic coding, seeking to identify both the manifest and latent messages (Cho and Lee 2014). The discourse analysis has led me to locate three types of appropriations of Mindfulness within a Jewish content. I have therefore crafted a three-part typology, which is, as any typology is, overly simplistic, but which I hope will help identify the various ways -most of the time cumulatively- in which Modern Orthodox Jewish educators are bringing Mindfulness within their own practice and teaching of Jewish religion.

The first approach I call the "Buberian" or "translation" approach: through this ethos, Jewish educators relate to Judaism in a new light, thanks to and *through* their experience of Mindfulness. The second approach I call the "cosmopolitan" or "cumulative" approach: via this ethos, Jewish educators bring together, without superposing them, the practices of Jewish religion *and* of Mindfulness meditation. The third approach I call the "biblicization" or "neo-traditional" approach: this type of strategy can be taken into two very different directions: either to deny all ideas of cultural import, or to better reappraise the Jewish tradition, by considering Jewish practice itself *as* a Mindfulness practice.

Through this three-fold analysis, I study Modern Orthodox appropriations of Mindfulness meditation as a case to better understand Postmodern religion in the context of cultural globalization: as a religious ethos unfolding in the paradoxical tension between traditional claims on the one hand, and overt cultural borrowing on the other—the later often used in the service of the former[29].

---

[28]　Mindfulness in the nineties was not yet available in Jewish contexts as widely as today.

[29]　French anthropologist Mary (2000) has observed such a process in the case of reappraisals of Pagan tradition while appropriating the Christian religion in Africa in the twentieth century.

## 2. Judaism "through" Mindfulness: The Translation Approach

In the lines below, I briefly retrace the process by which the concept of "Jewish Mindfulness" came to be through a series of cultural translations. But there is also another type of translation at play today, an ongoing one: the proponents of the "translation" stance argue that they gain better access to Jewish spirituality "through" Mindfulness. Mindfulness, they argue, has helped them see Jewish spiritual practice in a new light and is continuously enriching their practice and teaching of Jewish religion. I also call this approach the "Buberian approach", because Buber was one of the first to use comparative religion with a focus on Eastern spiritualities to read Judaism in a new light. For some of these teachers, secular Mindfulness becomes a tool in itself, as it helps them translate Jewish teachings in terms that are more meaningful and transformative.

### 2.1. From Buddhist Mindfulness to Jewish Mindfulness: A Journey of Cultural Translation

The genesis of Jewish Mindfulness is the outcome of a three-stage process of cultural translation. Cultural translation can be described as the process of translating—that is, of displacing, reframing and adapting a concept or a practice from a given cultural context to another. According to anthropologists of globalization, cultural translation has become a new ethos in the West, where "the comforts of tradition are fundamentally challenged by the imperative to forge a new self-interpretation based upon the responsibilities of cultural translations" (Robins 1991, p. 41).

Jewish Mindfulness was born out of such a "responsibility of cultural translation", as some rabbis, Liberal first, Orthodox later, have taken upon themselves the task to reinterpret their tradition through the lens of Mindfulness, in a new cultural context where Mindfulness practice has become so prevalent.

The symbolic journey of cultural translation from Buddhist Mindfulness to Jewish Mindfulness starts with the personal journey of young secular American Jews from the baby boom generation who went to learn meditation in India in the sixties. When they came back home in the early seventies, Joseph Goldstein, Sharon Salzberg, Jack Kornfield and Jacqueline Schwartz founded the first Western Buddhist school in America, the Insight Meditation Center (IMS). They became the first cultural translators of Vipassana Buddhism within Western culture: "Insight Meditation" is the English translation of the pāli word "*vipassana*", a meditation technique which gave its name to the Indo-Burmese school of Buddhism they had trained in (Fronsdal 1998, p. 165; Michaelson 2013, p. 16). One of the meditation techniques they focused on, "sati", had been rendered as "Mindfulness" by the first Western translators of the pāli canon at the end of the nineteenth century (Rhys Davis 1881; Bodhi 2011).

But in order to make meditation more available to a Western audience, they did not only linguistically translate the practice of Vipassana, they also translated it symbolically: as the name of their organization indicates, "Meditation Center", the "Buddhist" label was dropped, and replaced by the more neutral concept of "meditation." In terms of content too, they further secularized, westernized and psychologized the practice[30], leaving aside its Asian and Buddhist ritual/cultural/soteriological garments, to focus on the psychological this-worldly goal of well-being, highlighting three concepts which Westerners could all the more so relate to as they depicted them as universal:

> These three qualities—Mindfulness, compassion and wisdom—are not Burmese or Tibetan, Thai or Japanese, Eastern or Western. They do not belong to any religion but are qualities in our own minds and hearts, and many different practices enhance their growth.

By so doing, the first Western meditation teachers had started a trend that would later be called "Buddhism without belief" (Batchelor 1998). This was the first step of the Westernization and the secularization of Mindfulness in the West[31].

---

[30]　A process their own teachers had already started in Asia, see in (Fronsdal 1998, p.166; Olson 2005, p. 245).
[31]　The preliminary step having been in Asia: see previous note.

The second step happened when Western health professionals started using Mindfulness meditation outside of the field of spirituality, and inside the field of psychology. The first one to have done so, Jon Kabbat-Zinn, started integrating Mindfulness to his therapeutic protocol. When founding the Mindfulness Based Stress Reduction (MBSR) programs, the American psychiatrist took a step further in decontextualizing Mindfulness to recontextualize it within the context of Western secular therapeutic and well-being practice[32]. Science was the legitimizing agent that helped translate the practice, and therefore complete the integration of Mindfulness meditation within Western culture at large. As a consequence, today, "Vipassana-derived Mindfulness practices are taught in hospitals, clinics, prisons, schools, without any hint of their Buddhism source" (Fronsdal 1998, p. 165).

While Kabat-Zinn's definition of secular Mindfulness has become the reference for people who practice meditation outside of the field of Buddhism, it still strikingly parallels the IMS's:

> "Mindfulness is awareness that arises through paying attention, on purpose, in the present moment, non-judgmentally."[33]

The 'present moment', 'awareness' and 'non-judgment' seem to be the three main ingredients of a substantive definition of "Mindfulness", no matter the context. Then, it is the context which will give Mindfulness its flavor: Buddhist, secular, or ... Jewish.

Indeed, by making Mindfulness secular, Western Buddhist teachers and after them, scientists, have played the role of bridges between the Buddhist world, where Mindfulness came from, and the Western world, where it became enculturated, including within the Western religious field—Jewish and Christian mainly[34]—within which Mindfulness has slowly been integrated in the last three decades.

Another factor in the making of Jewish Mindfulness is the Jewish identity of these first Mindfulness teachers[35]; it is through one of his congregants, Sylvia Boorstein, a meditation teacher in the Western Insight Meditation school and a self-declared *Passionate Buddhist and faithful Jew* (Boorstein 1997), that conservative Rabbi Jonathan Slater started meditating. He then became a teacher of Mindfulness meditation himself, only within Jewish settings, such as the Manhattan Jewish Community Center (JCC). Most importantly, he wrote books on "Jewish Mindfulness" (Slater 2004, 2014), endeavoring to show how the ethos of Mindfulness that can be found within Jewish practice and texts. A couple of decades later, reconstructionist rabbi Sheila Peltz-Weinberg followed a similar process: she started going to IMS after the advice of one of her congregants who was a board member there (Peltz-Weinberg 2003, p. 99) and ended up becoming one of the major figures of the Jewish Mindfulness scene in America.

Together with Reconstructionist, Conservative and Renewal rabbis, in the early nineties, Rachel Cowan, Johanna Katz and Jeff Roth, Slater and Peltz-Weinberg founded the Institute for Jewish Spirituality (IJS): an organization dedicated to what was now called "Jewish Mindfulness", which was at the core of their idea of Jewish spirituality, as states the website home page:

> Discover ( ... ) spiritual practices like Jewish Mindfulness meditation or contemplative prayer to cultivate a new level of "aliveness"; feel more connected to yourself, others and God; access wisdom and resilience in stressful times; and find greater meaning in your life.[36]

In a Jewish context, Mindfulness meditation is described as one of the tools, next to "contemplative prayer", not only for better personal and interpersonal resilience[37] and meaning, but also for a

---

[32] On the impact of Kabat-Zinn's secularization of Mindfulness, see in (Michaelson 2013, p. 20).
[33] Jon Kabat-Zinn, "Defining Mindfulness", *Mindful.org*. Available at: https://www.mindful.org/jon-kabat-zinn-defining-Mindfulness/ (accessed on 27 April 2019).
[34] Mindfulness has also been imported in the field of Christianity (Tan 2011) and Islam (Thomas et al. 2016).
[35] Kornfield, Goldstein, Salzberg, Schwartz, Kabat-Zinn, and many more of their colleagues and students in the American Buddhist field are all Jewish, as has been noted by observers (Kamenetz 1994; Nisker 2003, p. 116; Lew 2001, p. 60) and by scholars (Prebish and Tanaka 1998, p. 3; Coleman 1999, pp. 77, 192).
[36] https://www.Jewishspirituality.org/ (accessed on 7 August 2019).
[37] On another page entitled "what are Jewish spiritual practices", Jewish Mindfulness comes, again, first, but is described less in terms of connection with God and more in terms of psychological well-being: "Mindfulness" would be is about

deeper connection with "God". While the rabbis who created Jewish Mindfulness were mostly from non-Orthodox denominations, they have been followed since then by Orthodox-ordained rabbis.

One of the first to have done so is American Rabbi James Maisels, who, after receiving Orthodox *smicha* (rabbinic ordination) at non-denominational Yeshiva Pardes in Jerusalem, founded a Jewish spirituality center, *Or Ha Lev* ("light of the Heart"), a decade ago. According to the organization's website,

> "The invitation of Judaism has always been to uncover the sacred in the mundane moments of life by paying attention. This is Jewish Mindfulness practice."[38]

In these two definitions of Jewish Mindfulness, we find both a core element of generic Mindfulness: "awareness", or "paying attention", as well as a core element of Jewish piety: a connection with the divine and finding the sacred in everyday life, Judaism being about the sanctification of life.

Hence, from the Buddhist Mindfulness that Goldstein, Kornfield and Salzberg were practicing in their orange robes in India in the sixties to the Jewish Mindfulness that kippa-wearing Jews are practicing today in America, there has been a three-stage process of cultural integration, which can be schematized as follows:

Buddhist Mindfulness- => Secular Mindfulness => Jewish Mindfulness

While the process of the integration of Modern Buddhism in the West has been analyzed as the "revenge of Benares", whereby Asian missionaries started reciprocating the "missionary thrust" that Westerners had directed towards them for centuries (Berger 1979, p. 164), it may be more accurate to say that this impetus was reciprocal: the integration of Mindfulness in America and the emergence of Jewish Mindfulness can be seen as the result of a mutual desire for cultural integration by Buddhist teachers on the one hand, and of cultural appropriation by Westerners on the other (Niculescu 2015a, 2015b; Sigalow 2019).

Yet the concept of cultural appropriation is to be used carefully, it bears the risk to imply an essentialist claim that there is a pure "original" cultural item which can be appropriated by another culture. This doesn't reflect the reality of intercultural contacts, which is more blurry: first, observers of the current context of the globalization of culture have highlighted a mutual play of influences between East and West (Cox 1977; Campbell 2015); second, as part of the cosmopolitanization of Western culture, cultural borrowing has become the norm (Turner 2011).

As such, Jewish Mindfulness teachers seem to be practicing a kind of cultural translation in order to reinterpret their own tradition. Jewish Mindfulness teachers started doing this "translation" work in the nineties, and this is what Orthodox rabbis are continuing today. And it works both ways: just as they are translating Mindfulness in Jewish terms, they are retranslating Jewish spiritual texts in terms of Mindfulness. Looking at Judaism in a new light through the practice of Mindfulness—this is what I call the Buberian approach.

### 2.2. Judaism through Mindfulness: A Mindful Lens to Rediscover Jewish Spirituality

"It is *through*[39] my experience in meditation that I have gained access to the spiritual consciousness taught by Rabbi Levi Yitzchak," states Jonathan Slater (2014, p. xxxi) in his reading of *Kedushat Levi*, a Modern Chassidic work by Rabbi Levi Yitzchak of Berditchev. For the American Conservative rabbi, one of the introducers of Jewish Mindfulness and author of a book on *Mindful Jewish living* (2004), it is his practice of secular Mindfulness (which he learned outside of the Jewish world) that has given him new lenses to see Chassidic teaching in a new light. Slater is among the first generation of Jewish Mindfulness teachers: he was Sylvia Boorstein's rabbi, and she trained him in Mindfulness meditation

---

"cultivating awareness and the ability to access wisdom and resilience." https://www.Jewishspirituality.org/about/what-are-Jewish-spiritual-practices/ (accessed on 7 August 2019).

[38]　https://www.orhalev.org/ (accessed on 7 August 2019).

[39]　My emphasis in the text.

([Niculescu 2015a](#)). Like him, other rabbis brought the practice of Mindfulness within their Jewish practice, because it gave them a new gaze on Jewish spirituality, especially on Hasidism. Rabbi Nancy Flam, in a recent collective book on Neo-Hasidism, shares an almost word-for-word observation: "I brought Mindfulness-meditation practice into my Jewish life, and read Hassidic texts through its lens. I encountered many texts that spoke of a kind of concentration similar to what I knew from deep Mindfulness meditation ( . . . )." (Flam 2019, p. 232).

Such is also today the vision of some in the new generation of Orthodox rabbis—mainly on the open-Modern Orthodox or non-denominational side: Ayal Robkin is an Orthodox-ordained rabbi and a day-school teacher at the non-denominational day school Heschel in New York. After receiving a classic Modern Orthodox education in the States, Robkin studied in Paris and received rabbinic ordination from rabbi Daniel Landes. Today, Robkin leads the "Mindfulness meditation minyan" that was mentioned in the opening of this article. Yet, he had rather negative prejudices about meditation and Mindfulness initially. But when he found himself learning for a year in Pardes, the non-denominational yeshiva in Jerusalem, he nevertheless decided to try. It was only in order to challenge his own views, and because he trusted the rabbi who taught it there (namely, James Maisels, whom we mentioned), that he decided to try Mindfulness Meditation. Progressively, he started appreciating the pragmatic wisdom of it[40]:

> I was learning at Pardes; and at Pardes I realized . . . I realized I was incredibly closed minded about certain things; especially things I thought about were . . . esoteric and hippie, and what not. And I decided to go to James[41]'meditation classes; and that ended up being its own challenge; I really loved it; until I moved to Chicago; and there's a teacher there named Sam Feinsmith; and, Sam was the first colleague of mine; and, we had discussions, about what Mindfulness is, and, I would sit in his class; and slowly I started realizing that Mindfulness was more grounded than I thought it was.

While he remains a critic of what he calls a 'hippie' side of meditation, it is the pragmatic, 'scientific' aspect of Mindfulness that convinces him, with the help of his own studies in psychology, which help him better understand how the mind works:

> That there was, more scientific rigor, behind the practice, of, meditation. Huh. That . . . there was certain parts of classic meditation teachings, that I found very . . . ethereal . . . that I was able to, over time, either go deeper into those things, or . . . for instance, understanding, the . . . so for instance, the classic Jon Kabat-Zinn Mindfulness, is being able to be present in the moment without judgment. And psychologically speaking, I found that to be really absurd, because . . . the idea of being present in the moment is psychologically very challenging, and, not judging things, is just not how that works in our minds . . .

> I studied a lot about psychology, as well . . . and . . . So I started looking into that, and I realized, 'oh, what he really means to say, is, 'try to notice what's happening to me in that moment and try to see what's arising in me, and try to notice the judgment.' And I was like 'oh, I was just being overly critical about something because I didn't understand it.' And over the time I think that's helped me develop my own Mindfulness practice, my own, teaching . . . to . . . . Really cater to high school students, and what they're going through . . . in this culture . . .

Robkin, an Orthodox-ordained rabbi working in a Jewish day school, feels safe teaching Mindfulness to his students: unlike the first Jewish Mindfulness teachers, who all sat in Western meditation centers, Robkin's exposure of Mindfulness meditation has not in any way brought him close to Buddhism:

---

[40] Interview on skype, Jerusalem-New York, 18 April 2019.
[41] Maisels.

> I've never had any concern that Mindfulness may be Eastern theologically: and I just never really cared about that; I never had any concerns about that; I never brought it in my teachings; and the thing is, I never really learned that anyway.

Like many of his contemporaries, he never learned Buddhist Mindfulness, but secular Mindfulness in a Jewish context: the Mindfulness techniques he learned were presented to him as "Jewish Mindfulness". And as a proponent of what I have called the Buberian approach, he has come to better understand the classical Jewish spiritual tradition of character refinement, or *Mussar*, through his approach of Mindfulness:

> Pretty much every single teaching that I have found, in the Buddhist world, I have found its corollary, in the Mussar world, or in the Hassidic world. Like the amount of things, that, you know, that like, Hassidic hints to, the illusory side of the self, which is a deeply Buddhist concept; I see it everywhere in Chassidus; and Mussar say the illusion of control; which is a very Buddhist idea; but you see it, about the concept of arrogance, in Mussar.

Indeed, like Buber, who saw parallels in Daoist and Zen teachings that helped shed light on Chassidic teachings[42] (Friedman 1976; Eber 1994), for Robkin, Mindfulness played the role of a magnifying lens: a transformative medium through which the traditional Jewish spiritual discipline of Mussar could suddenly be seen in a new light—so much so that he feels that he now needs one to teach the other: "you can't teach *Mussar* without Mindfulness and you can't' really teach Mindfulness without *Mussar*."

By seeing *Mussar* reflected in Mindfulness and Mindfulness in *Mussar*, Robkin embodies a "Hermeneutics of dialogue," which has also been called, in the context of teaching Jewish texts, a "Buberian approach" (Cohen 1999a, p. 49[43]). Kepnes (1992) described the Buberian approach as a four-step progression: "openness", "distanciation", "critical-historical methodology", and finally, "new insights". Robkin seems to have followed a similar route. Only he is not just dialoging with a "text," he is bringing two different texts together—the teachings of Mindfulness and the teachings of Torah. In order to read each in the light of the other, he also performs symbolic translation. He does so using what Jonathan Cohen, drawing from Michael Rosenack's work on educational translation, calls a "deliberative mode" of symbolic translation: a mode according to which the "world views, concepts, disciplines and cultural visions do not function as overarching norms, but rather as 'resources' drawn upon to the degree that they are seen to address problems' (Cohen 2003, p. 7). Just as Cohen predicts in the case of a Buberian approach to text, his detour/"distanciation" via "Mindfulness " causes him to have "new insights "(Cohen 1999b, p. 49) in his reading of Jewish texts.

Having rediscovered the meaning of traditional Jewish spirituality through his practice of Mindfulness, Robkin, who was so skeptical about Mindfulness, becomes a proponent of a cumulative approach: bringing Mindfulness meditation as part of the curriculum of Jewish schools, next to traditional Jewish religious practice—an approach that is also typical of the cosmopolitan stance.

## 3. Judaism "and" Mindfulness: A Cosmopolitan Approach

Up until the turn of the millennium, the "and" approach could be said to to be the domain of Renewal and liberal rabbis, who were not afraid to practice non-Jewish forms of spirituality next to their religion. Over the past decade, this cumulative or cosmopolitan stance has also been embraced by their Modern Orthodox peers: today, self-declared Halakhic Jews do not hesitate to also claim that

---

[42]  See also his other perspectives on the subject in the collective book "*Zen and Hasidism: the similarities between two spiritual disciplines* (Heifetz 1978).

[43]  In this important article, Jonathan Cohen was proposing four hermeneutic options to approaching Jewish texts, embodied in his view by four contemporary Jewish thinkers: the "hermeneutics of suspicion", embodied by Freud, the "hermeneutics of evolution", embodied by Fromm, the "hermeneutics of reverence", embodied by Strauss, and the "hermeneutics of dialogue", embodied by Buber.

they practice Mindfulness before Jewish prayer, or that their students in Orthodox day schools should do so. Having different feet in different cultures is not new for the Modern Orthodox. Thirty years ago, they were described as both "Cosmopolitans and Parochials" (Heilman and Cohen 1989). Today, it is Orthodox Jews who practice Mindfulness. Only the line seems to have become much wider.

*3.1. Mindfulness as a Spiritual Tool to Enhance the Experience of Jewish Prayer*

In a *Jewish Chronicle* article in which he reflects on the connection between Mindfulness meditation and Jewish piety, Samuel Landau, a young British Modern Orthodox rabbi holding a Doctorate in clinical Psychology, recounts how both practices came together for him during a *Yom Kippur* experience. He had come to the *bimah*[44] (oratory podium) to lead the prayer, finding himself upset and absent-minded, and he decided to use Mindfulness practice, as he had learned it at university during his psychology studies, to help him calm and focus so he could properly lead the congregation into Jewish prayer:

> I take a deep breath. I decide to relate to my thoughts differently and, remembering Mindfulness exercises, I breathe again ... The present moment becomes more full and rich ... I ground myself in the experience of the here and now and return to the text with self-compassion, without judgment, with Mindfulness.[45]

This discourse is very similar to the description of classic Buddhist Mindfulness technique, as this definition by Joseph Goldstein shows:

> "Mindfulness is the quality and power of mind that is aware of what's happening—without judgment and without interference."[46]

Landau, an Orthodox rabbi, is comfortable talking about "Mindfulness meditation", a technique he knows comes from Buddhism, because what he refers to is not the Buddhist Mindfulness Goldstein is teaching. The Mindfulness he refers to is secular Mindfulness: the scientific body/mind calming method inherited from Kabat-Zinn, which he learned at University during his studies in psychology.

But the decontextualization/recontextualization of Mindfulness is not the only reason that explains Landau's ease in using this foreign-originated meditation technique. It speaks to another phenomenon: the young Orthodox rabbi belongs to a Western culture that has become cosmopolitan. Cosmopolitanism has been described as a "mode of managing meaning" (Hannerz 1990, p. 238) in today's Western globalized societies, or even as a "virtue" characterized by "reflexivity with respect to other cultural values ideologies ( ... ) and hence acceptance of cultural hybridization" (Turner 2011, p. 253). This is why Landau's claim to use "Mindfulness" before leading Jewish religious services does not speak of cultural borrowing, and not even any more of cultural translation: it speaks of a cosmopolitan paradigm that he is part of. As a consequence, rather than "creolizing", he uses various practices side by side, each in its own time and for its own purposes, but each enhancing the experience of the other. Such is also the approach of Evan Mayse, who in an article in which he praises the importance of Halakha as part of a Neo-Hasidic theology, sees it side by side with the practice of Mindfulness:

> "The halakha of Neo-Hasidism is an empowering approach to Jewish life that incorporates daily practice, rigorous inner work, and constant spiritual exercise through Mindfulness and contemplation. Far from being an attenuated or denatured form of tradition, its core is the regular and disciplined performance of the mitzvot." (Mayse 2019, p. 166)

---

[44] The place where the *shaliach tzibur*, the prayer leader, will lead the assembly to prayer.
[45] https://www.thejc.com/Judaism/features/how-abraham-founded-the-biblical-school-of-Mindfulness-1.59791 (accessed on 27 April 2019).
[46] https://www.lionsroar.com/three-means-to-peace/ (accessed on 27 April 2019).

For Mayse and for Landau alike, the use of Mindfulness, far from threatening the practice of Jewish tradition, comes to introduce it. By so claiming, they show that Cosmopolitanism and cultural hybridity does not necessarily mean religious hybridity. Landau is not claiming to mix Mindfulness with Jewish practice, as is done for instance at IJS and at Or Ha Lev. He does not endeavor to read or translate Jewish rituals in the light of concepts and teachings coming from the world of Mindfulness. His approach, in this instance, is a cumulative one: he uses Mindfulness meditation as a preamble to Jewish prayer according to a functional articulation. His intention is to use Mindfulness as a pragmatic tool to calm his body/mind, not to blend two meaning systems. And indeed, he highlights the importance of drawing a symbolic boundary: "How do we square our Jewish sensitivities with a practice that is built on Buddhism, even in its Kabat-Zinn incarnation?"

By saying so, Landau shows, one, that he knows what he is borrowing from, and, two, that, as an Orthodox rabbi whose role is traditionally to delineate the zone of the permitted with the zone of the forbidden, he has to draw a line. And he does so by writing:

> The answer has to do with our approach; if we consider Mindfulness to be an end in itself, a state of being that is simply more beneficial for wellbeing, we may have missed an opportunity ... Rather, Mindfulness meditation should serve as a means to an end, the end being a deeper engagement with our Jewish journey, making Mindfulness a welcome addition to a Jewish life's toolbox.[47]

Landau seems to be positioning himself very much in the line of Rabbi Moshe Feinstein, one of the most famous twentieth century *ashkenaz* Halakhic authorities, when he ruled on the question to know whether American Jews could celebrate Thanksgiving—an American Holiday ([Berkowitz 2012], p. 219)—or not. To both of them, the criteria is the mindset of the practitioner. Hence, in the case of the use of Mindfulness meditation, for Landau, if the end goal of the practitioner is Jewish prayer, then it is fine. In an attempt to promote Jewish spirituality, he describes Jewish practice as the spiritual 'tool' with Mindfulness being just a sort of "can opener" meant to open the Jewish toolbox when it is closed.

Yet his choice of using Mindfulness before Jewish prayer, specifically in this instance, before the Yom Kippur prayer, which perhaps the highlight of the Jewish liturgical year, may seem surprising.

In particular, it may surprise any Jewish religious practitioner with a little acquaintence with Jewish spirituality, as Landau must be himself: that there is, indeed, a Jewish tool to help focus before prayer. It is described in a famous Talmudic passage[48] which recounts that the "first sages would 'sit' an hour" before prayer, so that they could "direct/focus their heads towards God".

The word used in the Hebrew text of this *mishna* is "*kavannah*", which means 'direction', 'focus', 'intention'. According to Orthodox Jewish authorities who are today attempting to reconstruct a proper tradition of "Jewish meditation", as we will see shortly, *kavannah* would even be "the most generic Hebrew term for meditation" (Kaplan 1985, p. 49). The same claim is expressed by Orthodox Rabbi Akiva Tatz in his Letters to a Buddhist Jew. While for Tatz, the real Jewish equivalent to meditation is prayer, he compares the Talmudic practice of Kavannah to what he considers to be "objectless meditation", which would be in his view the equivalent of the Zen meditation his interlocutor practices:

> "The mitzvah of tefilla, prayer, most intensely expressed what you know as meditation. There are closer forms too, such as the preparation for prayer, that more closely approximate what you call objectless meditation (the early chassidim—masters who transcend righteousness—used to spend an hour in preparatory meditation ( ... )." ([Tatz and Gottlieb 2004], p. 237)

---

[47] https://www.thejc.com/Judaism/features/how-abraham-founded-the-biblical-school-of-Mindfulness-1.59791 (accessed on 27 April 2019).
[48] Babylonian Talmud, Tractate *Brakhot* 32.B.

Why does Landau not use the concept of *Kavannah* here? Why does he not refer to the Talmudic verse, to say that he "stayed in silence to prepare his mind for prayer?" Why does he rather choose to use "Mindfulness"? The problem cannot be ignorance. Landau, as an Orthodox rabbi, would be aware of this Talmudic passage. The problem seems rather to be didactic: it is very likely that Landau, like generations of rabbis before him, and, like his contemporaries, has not learned to *use* Jewish spiritual techniques—even if he has *heard* about them or even *studied* them. He may very likely have learned the text, but not how to put it into practice. As a result, comes the shining simplicity, clarity and efficiency of "Mindfulness", transmitted with an impeccable pedagogy by non-parochial Mindfulness teachers in the very legitimate and culturally and religiously neutral field of University and psychology. In a recently published book on Neo-Hasidism, in which "mindfulness" seems to be a ubiquitous theme, contributors shared that it was the lack of actual teachers to guide them within the world of the Jewish meditative traditions that made them turn towards Mindfulness practice. Nancy Flam, a Female Reform rabbi and a Mindfulness meditation practitioner and teacher, writes:

> "I needed a teacher to show me how to practice, but there were significant obstacles. Most of those with an unbroken lineage ( . . . ) had been murdered in the holocaust. Furthermore, should I find such a living teacher, would he take a non-Orthodox woman as a student? So I stuck with Mindfulness meditation as my training towards emptiness, ayin (Nothingness) and non-self; and I brought that conditioning of mind into my daily engagement with Torah, 'avodah, and gemilut hassadim—with scripture, with service, and with works of compassion." (Flam 2019, p. 232)

While Flam tries to explain this phenomenon by the fact that she is a woman coming from a non-Orthodox world, Maisels, an Orthodox-ordained Male, relates a similar experience of barred access to traditional Jewish spirituality, for lack of pedagogic guidance:

> "I was transformed by the Piaczesner rebbe and consider him my rebbe, but for me it is complicated and profoundly sad that I was not able to learn this material from a living rebbe. ( . . . ) Instead I learned meditation as a living tradition and as part of an unbroken lineage from Western Mindfulness teachers (though no particular Western Mindfulness teacher has ever become my rebbe)." (Maisels 2019, p. 253)

This lack may be why Broder, Brodkin and their colleagues are today working at improving the Jewish religious didactics around Jewish spiritual and contemplative practices, as early as in Jewish schools including using Mindfulness in a Jewish context.

*3.2. Mindfulness in Jewish Schools: A Pedagogic Tool*

When he was in college, Sam Feinsmith, one of the first ordained "open Modern Orthodox" rabbis from Yeshivat *Chovevei Torah* in New York, discovered Mindfulness meditation. Sam had grown up in the world of Modern Orthodox day schools and Yeshivot in America, Israel, and back in America, in New Jersey, before feeling estranged from Orthodoxy as a teenager. During his college years at Columbia University and at the Jewish Theological Seminary (JTS), a conservative institution, he had started exploring other forms of spirituality—among which, various forms of Buddhist meditation practices that were available (mainly in the Zen and Tibetan tradition). While he has gotten back to an Orthodox observance since, he has maintained a serious practice of meditation, turning towards more secular practice of "Mindfulness". Today, he works for the Institute for Jewish Spirituality as the Program director of "Educating for a Spiritual life", focusing on bringing Mindfulness to day schools, especially Orthodox, where he knows from his own experience that the resistance is bigger but the need also greater. As a result, when he started teaching in Jewish day schools, he felt called to integrate the benefits of Mindfulness within his practice of Jewish religious teachings, so that it could benefit the students too. For him, just as for Landau, Mindfulness would help students find more *kavannah* (intention), as he puts it in Hebrew, for prayer:

All day schools are struggling with tefillah[49] (prayer). In the Orthodox world, the orientation of chiyuv (obligation) and matbeah (structure) can undermine kavvanah (intention), even though it provides a good structure. In addition, Orthodox youth are suffering from the same dysregulation and emotional stresses as the general population. I have spoken with administrators at Orthodox day schools who have implemented secular Mindfulness programs to support youth with these challenges. But they have no idea how to locate the practices in a Jewish framework. I wanted to give Orthodox educators tools—what I can kavvanah skills—to help them infuse tefillah with more personal meaning and also transform tefillah into a practice for helping students to cultivate authentic spirituality as a foundation for thriving (*a la* Lisa Miller's work in *The Spiritual Child*). In short, as an alternative to "Do tefillah because halakha says we should do it," I wanted Orthodox educators to be able to say to kids, "Let's do tefillah together because it will help us grow into the people we hope to be.

For Feinsmith, the need for Mindfulness in Jewish day schools—in the Orthodox ones in particular—is twofold. First, like everywhere else, it speaks to mental health needs, which are a societal issue today; second, and this is specific to Jewish Orthodoxy, it speaks to the downsides of this type of education: "obligation", notes the open Modern Orthodox rabbi, can get in the way of intention; duty, can get in the of meaning. Feinsmith's endeavor to bring Mindfulness to Orthodox Jewish school is based on his own desire to provide Jewish observant kids today with what he may himself have lacked in his own Jewish education: learning Jewish religion as a spiritual practice, and as a path of self-development—helping students to become the best people they can.

For rabbi Hillel Broder, it is also a spiritual issue. "Z lot of prayer education has to do with learning the language and choreography", he says, "but not the spirit of the prayers[50]."

This is why, in his application for the Kohelet prize for Jewish education, he wrote: "our challenge, as educators, is to teach tefila so that it is not only a normalizing, communal practice but a spiritual one–one that will nurture and sustain our students' rich emotional, religious, and yes, spiritual lives as they mature through and beyond high school."

He uses Mindfulness in the service of Jewish prayer as a preparatory tool for students to be focused and present in order to be able to receive inspiration for Jewish prayer:

> "In the particular capacity of a facilitator of an alternatively structured tefila, I have built a tefila space that both quiets and focuses students around conventional tefila practice through deliberate practices of Mindfulness meditation built into the tefila–before, and during the prayers themselves–while still maintaining the full ritualized practice of Orthodox prayer. My theory, which has played out for the past four years and for over 100 students, is that a stilled body allows for a focused mind, and a quieted, focused mind allows for optimal tefila experiences. Setting students up for success, in other words, has everything to do with setting up tefila properly so that it might be a strengthening, cathartic, and transformative experience."[51]

Brody's success is, however, not yet widespread in the world of Modern Orthodox Schools in America. Since he recently became the director of a new program offered by the Institute for Jewish Spirituality, "Educating for a Jewish Spiritual Life, program for Jewish day and supplementary school educators and students[52]," a program "designed to serve the entire Jewish community, including

---

[49] The transliterations for the words "tefilah," prayer, and "kavvannah," intention, varies. I reproduce it as written by the authors.

[50] Ben Harris "Minyans for meditation, artists and doubters", *precit*.

[51] Hillel Broder: "Building a Listening Room, Maturing Student Prayer: Creating Intentional Religious Environments and Practices in the Traditional Jewish Day School", *Kohelet Prize Database*, available at: https://koheletprize.org/database/building-listening-room-maturing-student-prayer-creating-intentional-religious-environments-practices-traditional-Jewish-day-school/ (accessed on 1 December 2019).

[52] https://www.Jewishspirituality.org/teach-others/educating-for-a-Jewish-spiritual-life/ (accessed on 7 August 2019).

Orthodox schools", and which "focused on meditation skills that then get applied in *middot* work and *tefillah* in a non-dogmatic, experience-based fashion that educators can adapt to their setting," Feinsmith was able to start implementing his vision. However, while he encountered a very positive reception in all the other schools, he notes: "no Orthodox day schools have signed up yet, though I am hoping they will".

There seems to be two reasons for that: The first is the challenge of integrating a new practice coming from the outside, within a Jewish setting, and the second is the absence of a culture of spirituality as "work on the self to transform the self (Foucault 2001, p. 16). within the Orthodox world.

"I have spoken with administrators at Orthodox day schools who have implemented secular Mindfulness programs to support youth with these challenges", says Feinsmith, "but they have no idea how to locate the practices in a Jewish framework." The problem does not seem to be about the Buddhist origin of Mindfulness. It seems to be more structural:

> They are not struggling to justify (it), because the practices they are doing are totally secular and don't threaten their Judaism in any way. The issue, I believe, is that the educators themselves have thin spiritual lives and don't know how to guide the students in an exploration of the inner life. There is no sense that kavvanah is a muscle that can be built. ( . . . ) The resistance here is more about finding the instructional time, asking people to shift their mindset, etc."

Indeed, traditionally raised Orthodox educators who have not ventured outside of the cultural boundaries of Judaism may very well not be aware of the Buddhist origin of Mindfulness. So the problem of taking in what they simply see as a secular therapeutic practice would simply be because it would be a new practice, which they would have to fit into the traditional Jewish curriculum, and a practice calling for a new culture of spirituality which they, according to Feinsmith, mostly lack: spirituality as self-awareness and self-development rather than spirituality as fixed liturgic practices and texts. This distinction is what Heelas and Woodhead have called the difference between "spirituality" (which they define as "subjective life"), and "religion", which they define as "life as") (Heelas and Woodhead 2005, p. 3). In their view, religion is currently being supplanted by spirituality or "subjective life", in the process of what they call the current *Spiritual revolution* in the West.

And it seems like it would still take a revolution, according to Feinsmith, in Orthodox day school, to understand that "*kavannah*" is not a concept but a practice: a skill to develop, "a muscle to be built."

Still, just like Landau, it is "Mindfulness", and not the specific Jewish concepts for spiritual practice: "*kavannah*" or "*mussar*", that he brings in. This speaks, again, of the difficulty of recreating a proper Jewish pedagogy of spiritual practice. Such a trend is nevertheless emerging progressively, particularly in Israel, where in some Orthodox Yeshivot, rabbis are focusing on formally teaching the Kabbalistic practice of *Kavannot* as a meditation technique (Guzmen-Carmeli and Rubin 2014).

## 4. Judaism "as" Mindfulness: Competitive and Integrative Approaches

The tenants of the competitive approach opt for what Berkowitz has called the "biblicisation strategy" (2002: 144): they argue that Jews do not need to look for Mindfulness elsewhere, as Judaism is already mindful. The tenants of the integrative approach do not deny the cultural import. Rather they integrate it as part of their process of reappraisal of Jewish meditation. In both cases, the creative result has been, over the past half-decade, the retrieval and reconstruction of a tradition of 'Jewish meditation', mostly thanks to the rediscovery and the publication of texts of traditional Jewish meditation—mostly Hasidic, but also Kabbalistic. The rabbis who have knowledge of both Mindfulness and Jewish meditative texts describe Jewish meditation *as* mindful and proceed to works of symbolic translations, which are typical of Postmodern religiosity in today's global context.

### 4.1. The Biblicization Strategy: From Denying Cultural Import to Competitive Creativity

In the last lines of his article, after having spoken of the wonders of Mindfulness meditation as a preamble to Jewish prayer and after having specified how it is permissible for Jews, rabbi Landau seemingly wonders: "does Judaism have an analogue?" He is not really wondering. His question is just rhetorical, and only means to introduce his next argument, which is also the main point of his article, as indicates the title: "how Abraham founded the biblical School of Mindfulness". Using the biblical passage of the almost-sacrifice of Isaac by his father Abraham (the *Akedah*), Landau intends to prove that the Torah contains its own built-in practice of Mindfulness. To him, Abraham saying to God "*hineini*" ("here I am") is an ideal-typic mindful stance:

> The Akedah passage may suggest that a mindful or Hineini approach is part of the Jewish experience; a way to engage one's essence meaningfully ( . . . ).[53]

This type of argument is not new. In fact, it reflects a very old rabbinic tradition, which Beth Berkowitz calls the "nativization" or "biblicization" strategy. It is built on the Talmudic "*Ketivta*" principle, which consists in saying: "since it is written in the Torah, we do not derive it from them[54]" (Berkowitz 2012, pp. 144, 159). To this end, rabbis seek hints, words, or situations found in Jewish texts in order to claim that a foreign practice that is being imitated is *not* foriegn, because it has always been Jewish. In short, by saying "we have it too", they can de facto import a foreign practice, while de jure denying any type of cultural appropriation.

When the "we have it too" becomes "we had it first", the biblicization strategy goes one step further. A startling example, which touches upon our topic here, is a widespread opinion in the Orthodox world, according to which all Eastern spiritualities would be, in fact, coming from Judaism. Proponents of this competitive claim bring the Torah as a textual proof: using a medieval commentary on a biblical verse mentioning Abraham's gifts to the "East", according to which these were "spiritual gifts[55]," outreach rabbis today draw "evidence" that Eastern spiritualities derive from the Torah. The programmatic conclusion of this rhetoric is that therefore, Jews should not look elsewhere to quench their spiritual thirst, since they belong to the "source": "in fact", writes Orthodox Rabbi Akiva Tatz in his *Letters to a Buddhist Jew*, "our sources clearly indicate that the wisdom of the East is built on principles of spiritual teaching that Abraham sent to those parts[56] as gifts to his sons" (Tatz and Gottlieb 2004, p. 45).

By doing so, Tatz exemplifies the type of biblicization argument about which Berkowitz writes "rabbis can create a loophole by which a copied custom can be considered native, if one's hermeneutical skills are up to the task" (Ibid, p. 145). And in his article on "The biblical school of Mindfulness", rabbi Landau seems "up to the task": in it, he uses cultural or symbolic translation to read Abraham's declaration to God "here I am" as a declaration of Mindfulness:

> Avraham seems to understand that this interaction with the Divine will require a certain mode of being, a mindful mode. [ . . . ] And so Abraham answers "Hineini"—Here I am, in my totality, in the present moment, non-judgmentally accepting my experience.[57]

This definition of a Jewish mode of Mindfulness is strikingly similar to the definition of Buddhist Mindfulness displayed on the Insight Meditation Center (IMS) website:

---

[53] Landau, *precit*.
[54] Babylonian Talmud, Sanhedrin, 52 b.
[55] Rashi on Genesis 25.6.
[56] Of the world: India and Asia in general.
[57] Samuel Landau, "How Abraham founded the biblical school of Mindfulness", *The Jewish Chronicle*, October 8, 2015. Available online: https://www.thejc.com/Judaism/features/how-abraham-founded-the-biblical-school-of-Mindfulness-1. 59791 (accessed on 27 April 2019).

At the heart of Insight Meditation is the practice of Mindfulness, a practice of moment to moment observation which cultivates a clear, stable and non-judgmental awareness.[58]

The parallels can be seen not only in the description of the experience, but also in the terminology used, with similar words or concepts such as "moment" or "non-judgment". Knowing that Landau is familiar with Insight Meditation Mindfulness, as he opens his article talking about this practice, it seems clear that his "biblical Mindfulness" takes ground on the Buddhist-derived secular definition of Mindfulness. The only difference is the backdrop of the experience. While secular Mindfulness invites people to be present to life as it is, Jewish Mindfulness invites people to be present to God as infusing life as it is. The backdrop in Jewish Mindfulness is an encounter with the divine.

Such endeavors of appropriations, in the Orthodox world, of "Mindfulness" as an "already Jewish" ethos are typical of what Ahad Ha'am has called "Competitive imitation" (Ahad Ha'am 1912):

They could not rest satisfied until they found an ancient legend to the effect that Socrates and Plato learned their philosophy from the prophets, and that the whole of Greek philosophy was stolen from Jewish books [ … ].

Elsewhere, I have called this type of strategy the "Buitoni Strategy," in reference to a popular commercial in the nineties. The television spot was showing a wife running after her husband who had jumped on a boat full of cans of his favorite ravioli, and shouting: "come back, honey! I have the same at home!" (Niculescu 2014).

This type of argument has been particularly used by *Chabad*, as early as in the seventies: witnessing the number of Jewish baby boomers, who, in the midst of the Hippie wave, were flocking towards Hindu ashrams and Buddhist meditation centers, some Orthodox rabbis in America felt alarmed, fearing the acceleration of an assimilation process that was threatening the "Jewish continuity"(Linzer 1996). The last rabbi of Lubavitch, Menachem Mendel Schneerson in particular, quickly responded counter-competitively, through outreach (*kiruv*) action. Believing in emulation rather than arguments and in seduction rather than guilt, he sent out charismatic emissaries whose task was to re-inspire young disaffiliated Jews among a generation of "spiritual seekers" (Roof 1993) by giving them a taste of spirituality, within the frame of their tradition. The two champions of this endeavor, Shlomo Carlebach and Zalman Schachter-Shalomi, ended up each starting their own trend; the first created Jewish spiritual songs he played on American campuses in folk music festivals, and the second ignited a 'paradigm shift' (Singer 2000) by infusing Jewish rituals with Eastern contemplative practices.

Both proved very successful (Ariel 2003): Carlebach, whose prayer tunes are sung today in most *Ashkenazi* synagogues all around the global Jewish world, brought back thousands of Jewish hippies to Orthodox Judaism (Zeller 2006); and Zalman's 'Jewish Renewal' movement has since then spread all over the Jewish diaspora and in Israel (Magid 2005). Another outreach (*kiruv*) entrepreneur has had a lasting impact on the contemporary landscape of Jewish piety: by reconstructing a tradition of Jewish meditation from biblical sources, Orthodox rabbi Aryeh Kaplan used the "biblicization strategy" as a creative process.

### 4.2. Symbolic Translation: Reconstructing Jewish Meditation to Theorizing Mindfulness as Neo-Hasidism

*Jewish Meditation* (1985), Kaplan's most famous and best seller book, giving an overview of "traditional Jewish meditation techniques," is the second of a three-part series on Jewish meditation[59]. This endeavor of giving access to a mainstream audience to Jewish mystical and contemplative practices has been considered as a direct outcome of the Lubavitcher Rebbes' instructions (Ophir 2013). Kaplan

---

[58] "Introduction to Mindfulness meditation", Insight Meditation Center. Available online: https://www.insightmeditationcenter.org/2016/07/introduction-to-Mindfulness-meditation-3/ (accessed on 27 April 2019).
[59] It's been preceded by *Meditation and the bible* (1978) and followed by *Meditation and Kabbalah* (1986).

is not hiding his counter-competitive agenda either in writing this book. In fact, he is very upfront about it in his introduction (Kaplan 1985, p. viii):

> Today, many American Jews have become involved in Eastern religions. It is estimated that as many as 75 percent of the devotees in some ashrams are Jewish, and large percentages follow disciplines such as Transcendental Meditation.

> When I speak to these Jews and ask them why they are exploring other religions instead of their own, they answer that they know of nothing deep or spiritually satisfying in Judaism. When I tell them there is a strong tradition of meditation and mysticism, not only in Judaism, but in mainstream Judaism, they look at me askance. Until Jews become aware of the spiritual richness of their own tradition, it is understandable that they will search in other pastures.

In this book, Kaplan resorts to "textual archeology", as he endeavors to find various biblical sources to prove a tradition of Jewish meditation. Yet his "retrieval" of a terminology of Jewish meditation sometimes seems more like a construction than like a reconstruction: most of the equivalences he is proposing—for instance arguing that Yitzhak's "*siyach*" (conversation) in the fields, is an instance of Jewish meditation in the bible (Kaplan 1978, p. 103)—are not entirely convincing. This is also Alan Brill's opinion, who compares Kaplan's symbolic translation of Hebrew roots into Modern meditation terminology, to the work swami Vivekananda, one of the introducers of Hindu meditation in the West, did when he took "the medieval Kurma Purana and turned it into Modern meditation about sitting straight and focusing[60]."

Tomer Persico, who himself recently authored a book offering both a chronological and typological perspective of "Jewish meditation" (Persico 2016), makes similar observations when analyzing the discourse of Rabbi Israël Besançon. To him, it is clear that the French Bratslav teacher, who teaches in Tel Aviv, "is influenced by the contemporary spirituality scene, and within it notably by Modern Buddhist *Vipassana* meditation, whose model of practice he uses in order to construct his own."

Not all of Persico's arguments in his critique are convincing, for instance when he equates Besançon's call for God to "mend our errors and see in a clearer way" (ibid., p. 57)" with the "*Vipassana*'s aim, which is to develop a "right vision." Still, the general argument makes a point, as the trend is real. Many Orthodox spiritual teachers today in *kiruv* Yeshivot in Israel or in the United States are either *Baalei tshuvah*—returnees to Jewish observance—often after having explored Eastern spiritualities themselves. Or, these teachers are, at least acquainted, through personal reading, or through school or University studies, with Buddhist meditation or Mindfulness books, which are now very much a part of the Western intellectual landscape.

This is why some of them culturally translate Jewish spiritual concepts into the terminology of Mindfulness. Indeed, most English translation of Jewish spiritual works today, especially Chassidic work, translate various Hebrew terms with the concept "Mindfulness." For American Modern Orthodox rabbi and psychologist Ben Epstein, for instance, "Mindfulness" translates symbolically as "*yishuv hada'at*" (literally the "establishing of knowledge")[61]. Likewise, most of today's English editions of "*bnei machshava tova*" (literally as "the sons of the good thinking"), a Modern Chassidic spiritual treatise written by rabbi, the late rabbi of the Warsaw ghetto, use the term "Mindfulness" to Shapiro's concept: in his translation of the book title, Yaakov Shulman says that it means "literally the society for positive Mindfulness" (Shulman 2017, p. i). Like Epstein, he translates both "*machshava*" (thinking) and "*da'at'*" (knowledge) as "Mindfulness" (ibid., pp. 9–10). Similarly, in her translation of the same book, Andrea Cohen Kiener uses the term "Mindfulness" to render the Hebrew expression:

---

[60]  Alan Brill. "Rabbi Aryeh Kaplan. Creating 20th century meditation, Blog "The book of doctrines and opinions. Notes on Jewish theology and spirituality" *Kavannah*. https://kavvanah.wordpress.com/2018/10/09/rabbi-aryeh-kaplan-creating-20th-century-Jewish-meditation/ (accessed on 12 August 2019).

[61]  Ben Epstein "Be. Here. Now. An introduction to Jewish Mindfulness", October 17, 2018, available at: https://www.jpost.com/Magazine/Be-Here-Now-569660 (accessed on 27 April 2019).

שכלול שטוח וחזוק המחשבה -literally a « wide intellect and a powerful thinking » (Kiener 1999, p. 12).

The English word, and the concept of "Mindfulness", has especially become central not only in translating Modern Hasidic works, but also in the current crafting of a "neo-Hasidic" theology. In a recently published book about neo-Hasidism, Art Green, one of the editors, uses a terminology to speak about two major Kabbalistic concepts, *da'at* and *binah*, which is very close to the Mindfulness vocabulary: "Da't, best translated as "mind" or "awareness," indeed resides within ( . . . ) the reduced consciousness of our ordinary mental self. This is the ordinary conscious self. But human beings are capable of insight that comes from a more profound realm of existence. It is called binah ( . . . ) "understanding of the heart" (Green 2019, pp. 21–22).

In the same volume, James Maisels goes further, when in an article unequivocally entitled "Neo-Hasidic Meditation: Mindfulness as a Neo-Hasidic practice," he claims that the Piaseczner rebbe "present a systematic and powerful path of Jewish Mindfulness" (2019, p. 254). The reason for him is a very strong conceptual mirroring, which makes him see these Modern Hasidic teachings *as* Mindfulness instructions:

> "Those familiar with Buddhist-based Mindfulness teachings as transmitted in the West and its practice will notice that there are very few gaps between Western Mindfulness teachings, the Hasidic goals, and the Piaseczner's particular practice as I have presented it above. Perhaps this is why my Mindfulness practice, which is profoundly based on Western Buddhist Mindfulness practice and theory, has always felt so fully integrated with and supportive of my broader Jewish practice.". (Maisels 2019, p. 258)

All the while doing so, he does not, in contrast to the proponents of the biblicization strategy, deny the cultural import. On the contrary, he highlights it—all the while acknowledging the differences and "tensions" that this can sometimes create (p. 260). But mostly, he credits Mindfulness for making his Jewish spiritual practice deeper and clearer: "Perhaps my explicit acknowledgment of the import and impact of these outside sources makes my practice neo-Hasidic neo Hasidism recognizes with gratitude wisdom from other spiritual traditions that enriches our own spiritual practice." (Maisels 2019, p. 258).

These types of mirroring stances are at the core of the process of cultural translation. Especially when, as Prothero noted in his study on one of the first Western Buddhists, founder of the Theosophical movement, Colonel Henry Olcott, a vocabulary from a given culture is articulated within the grammar of another. Prothero uses linguistic theory to explain this process: "individuals", he writes, "seem to be almost as insistent about clinging to inherited grammatical forms as they are comfortable with adopting new vocabularies" (Prothero 1996, p. 8).

About Olcott and his "Protestant Buddhism", Prothero writes that he created "a creolization of the Protestantism of his youth and the Buddhism of his adulthood." As a result, "while the lexicon of his faith was almost entirely Buddhist, its grammar was largely protestant" (Ibid., p. 9).

Transposing this analysis to the case of "Jewish Mindfulness", we can say that Orthodox rabbis who have adopted the concept of Mindfulness are using a "Mindfulness vocabulary" within the context of their "Jewish grammar." The result of this cultural appropriation, for the Modern Orthodox rabbis who are proponents of Jewish Mindfulness, is an enriched appraisal of the Jewish tradition.

## 5. Conclusions: Mindfulness and the American Way into Neo-Hasidism

In this article, I have looked at the uses of Mindfulness by Orthodox Rabbis and educators to enrich their religious practice and teaching, as a case to study Jewish institutional strategies of cultural adaptation in a global context. Focusing on the encounter with "Mindfulness", a Buddhist-derived secular Meditation practice which has grown within Western culture at large, I have noticed three forms of reappraisal of Jewish Spirituality thanks to this encounter: *through* the practice of Mindfulness (a translation approach), *alongside* it (the cumulative approach), or by looking at Jewish practice *as* a mindfulness practice (a neo-traditional approach). Not only are these approaches not mutually

exclusive, but, most of the time, they are superposed: individuals move between the "through", "and" and "as" stances.

In all these instances, however, Mindfulness plays a similar role: that of spiritual didactics. It is a conceptual and practical tool that, according to the actors of this emerging field of "Jewish Mindfulness" which often self-describes as "Neo-Hasidic," helps enhance the experience and the teaching Jewish religiosity as a spiritual practice.

A few decades ago, such discourses would have been unthinkable. The integration of Mindfulness within contemporary Jewish religious teaching in the Orthodox world thus exemplifies the stance of "adapted acculturation" Cohen and Heilman were talking about when defining the stance of the Modern Orthodox as "Cosmopolitan and Parochials" (1989). By doing so, these Modern Orthodox rabbis, who are Mindfulness practitioners and teachers, are showing that boundary fluidity may be the best strategy to reaffirm a group's symbolic boundaries. While most of them come, like the concept of "Jewish Mindfulness", from America, the phenomenon of the integration of Mindfulness within Jewish practice is becoming a global phenomenon: today, not only in Israel and in the English-speaking countries of the Jewish diaspora (England, Canada, Australia), but also in other Western Jewish communities (Berlin[62] or Buenos-Aires[63]) where the influence of the Americanized Jewish global culture has extended, local communities are teaching "Jewish Mindfulness". Modern Hasidism, as part of other Modern Jewish Movements such as Reform and Orthodox Judaism, was born in Eastern Europe. Since the seventies, the emergence of a Postmodern form of Hassidism, which calls itself neo-Hasidic theology, has been crafted by American rabbis (from Jewish Renewal and Liberal backgrounds first, and in the past half-decade, from Modern Orthodox ones). Today, this movement is also unfolding in Israel. This tends to indicate that since the aftermath of World War II, Judaism's creative hub has followed the trends of Jewish migrations and has glided from Poland and Germany to the United States, and now to Israel as well.

Ahad Ha'am probably had not heard about "Mindfulness" when he wrote about "competitive imitation." At the time of his writing, the term was barely starting to be circulated by the first translators of the pāli canon. But he was talking about a far-reaching pattern in the trajectory of the construction of Jewish culture—of which, we see but a new example today, through the encounter with Buddhism and its secular applications in Western culture. And as far as the pattern goes, it seems that he was right: by appropriating "Mindfulness" and making it a new Jewish religious ethos, once again in their historical trajectory, Jewish educators are showing creative ways to renew the Jewish Experience and to make it more attuned to the spiritual yearnings of their peers. Is that not, in the end, the genius of cultural resistance?

**Funding:** This research received no external funding.

**Acknowledgments:** The author wants to warmly thank Joshua Jacobs and Avery Robinson for first draft English language editing, and very warmly Akiva Daube for final version English language editing-twice.

**Conflicts of Interest:** The author declares no conflict of interest.

## Primary Sources

Boorstein, Sylvia. 1997. *That's Funny, You Don't Look Buddhist*. On Being a Faithful Jew and a Passionate Buddhist. San Francisco: Harper.

Epstein, Ben. 2019. *Living in the Presence: A Jewish Mindfulness Guide for Everyday Life.* Jerusalem: Urim Publications.

---

[62] "Ruach haShabbat. Jewish Mindfulness and Meditation in Berlin. Available at: https://www.facebook.com/events/262946844608870/ (accessed on 1 December 2019).

[63] Mishkan a Jewish Spirituality Center. Available at: https://www.bh.org.il/mishkan-jewish-spirituality-center-buenos-aires/ (accessed on 1 December 2019).

Flam, Nancy. 2019. Training the Heart and Mind towards expansive Awarness: A neo Hasidic journey. In *A New Hasidism. Branches*. Directed by Arthur Green and Ariel Evan Mayse. Philadelphia: The Jewish Publication Society, pp. 223–250.

Green, Art. 2019. A neo-Hasidic Credo. In *A New Hasidism. Branches*. Directed by Arthur Green and Ariel Evan Mayse. Philadelphia: The Jewish Publication Society, pp. 11–40.

Harris, Ben. Minyans for meditation, artists and doubters: How Jewish day schools are reimagining daily prayer. The Jewish Telegraphic Agency (JTA), September 25, 2019. Available online: https://www.jta.org/2019/09/25/united-states/minyans-for-meditation-artists-and-doubters-how-Jewish-day-schools-are-reimagining-daily-prayer (accessed on 1 December 2019).

Heifetz, Harold. 1978. Zen and Hasidism: The Similarities between Two Spiritual Disciplines. Hoboken: Ktav Publishing.

Kaplan, Aryeh. 1978. *Meditation and the Bible*. New York: Weiser books.

Kaplan, Aryeh, 1985. *Jewish Meditation*. New York: Schocken.

Maisels Jacobson, James. 2019. Neo-Hasidic Meditation: Mindfulness as a Neo-Hasidic practice. In *A New Hasidism. Branches*. Directed by Arthur Green and Ariel Evan Mayse. Philadelphia: The Jewish Publication Society, pp. 251–270.

Mayse, Ariel Evan. 2019. Neo Hasidism and *Halakha*: The Duties of Intimacy and the Law of the Heart. In *A New Hasidism. Branches*. Directed by Arthur Green and Ariel Evan Mayse. Philadelphia: The Jewish Publication Society, pp. 155–222.

Michaelson, Jay. 2007. *God in Your Body: Kabbalah, Mindfulness and embodied Spiritual practice*. San Francisco: Jewish lights.

Michaelson, Jay. 2013. The Man who taught the World to meditate. *Huffington Post*, June 30. Available online: https://www.huffpost.com/entry/sn-goenka-dead_b_4016374 (accessed on 1 December 2019).

Nisker, Wes. 2003. *The big bang, the Buddha, and the baby-boom; the spiritual experiments of my generation*. San Francisco: Harper San Francisco.

Ophir, Nathan. 2013. The Lubavitcher Rebbe's Call for a Scientific Non-Hasidic Meditation. *B'or Ha'Torah* 22: 109–123.

Peltz-Weinberg, Sheila. 2003. The Impact of Buddhism. In *Beside Still Waters: Jews, Christians, and the Way of the Buddha*. Directed by Harold Kasimow, John P. Keenan and Linda Klepinger Keenan. Boston: Wisdom Publications, pp. 99–113.

Persico, Tomer. 2016. *Jewish Meditation. Contemporary Development of Spiritual Jewish Practice*. Tel Aviv: Tel Aviv University Press (Hebrew).

Ryan, Tim. 2012. *A Mindful Nation*. New York: Hay House.

Shapira, Kalonymus Kalman. 1999. *Conscious Community: A Guide to Inner Work*. Translated by Andrea Cohen Kiener. New York: Jason Aronson.

Shaw, Marvin. 2017. *Mindful Judaism: A Jewish Guide to Beating Stress and Anxiety*. Manchester: i2i Publishing.

Slater, Jonathan. 2004. *Mindful Jewish Living: Compassionate Practice*. New York: Aviv Press.

Slater, Jonathan. 2014. *A Partner in Holiness. Vol 1. Deepening Mindfulness, Practicing Compassion and Enriching Our Lives through the Wisdom of R Levi Yitzhak of Berditchev's Kedushat Levi*. Woodstock: Jewish Lights.

Shapira, Kalonymus Kalman. 1999.(translation Andrea Cohen Kiener) 1999. *Conscious Community: A Guide to Inner Work*. New York: Jason Aronson.

Shapira, Kalonymus Kalman. 2017. (translation Yaakov David Shulman) 2017. *Experiencing the Divine. A guide to Jewish Spiritual Practice and Community*. Jerusalem: dotletterword.co.

Tang, Chade-Meng. 2012. *Search inside Yourself*. New York: Harper.

Tatz, Akiva, and David Gottlieb. 2004. *Letters to a Buddhist Jew*. Southfield, MI: Targum Press.

Zeller, David. 2006. *The Soul of the Story: Meetings with Remarkable People*. Woodstock: Jewish Lights.

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
