# Peer review of "“Jewish Mindfulness” as Spiritual Didactics Teaching Orthodox Jewish Religion through Mindfulness Meditation"

_religions, doi:10.3390/rel11010011_

Round 1
Reviewer 1 Report
My recommendation is to accept the article for publication pending on some—a bit more than "minor," but not “major” either—revisions.
I offer some general comments, which I think should be corrected:
There is no methodological chapter (separate and clear, as required by the journal) that explains what was done for the study: what is the method of research? What texts were collected, and how? How was the content analysis carried out? Why were these texts were chosen and not others? How many texts were reviewed? Who are the authors of the texts quoted in the article, beyond their religious affiliation and role, something about their religious background that would tell us why they chose to adopt mindfulness meditation for their Jewish religious toolbox? And so on. How can one determine that the use of mindfulness meditation among Orthodox Jews is a widespread phenomenon or is gaining momentum? A reference to how often the term appears in forums and Jewish sites could be of help here. It is a test that can be quickly done using technical tools like Google Trends. We must be provided with more information apart from quotes from texts (that it is not clear to us why these were specifically selected). Another matter regarding methodology: The article mentions interviews conducted for the research. Details must be provided. The article may have been written some time ago. I suggest that the author refer to and expand on Emily Sigalow's work, especially her new book: American JewBu: Jews, Buddhists, and Religious Change. The book describes the process of mutual fertilization that exists in the US between Buddhism and Judaism and addresses a significant portion of the points that are also raised in the article. How, if anything, does Orthodox writing about mindfulness meditation relate to spiritual practices similar to meditation in Judaism? Furthermore, what is Jewish meditation? The author can find help on this topic in Tomer Persico's excellent book on Jewish meditation (unfortunately currently only published in Hebrew), in Chava Weissler's writings about the activities of the US Kabala Center, or in ethnographic works by Giller, and by Rubin and Guzmen on prayers of intent as a meditative act among kabbalists. The article is somewhat weak in theory, although it raises a very intriguing claim that not only does the practice of mindfulness meditation among Orthodox Jews not weaken their faith, but it actually strengthens it and their religious-Jewish affiliation. Are there any similar instances (at present. The examples presented are historical and there too references to the literature are lacking) where a religious system borrows ideas or tools from outside for its creation? For example, writing about religious fundamentalism clearly explains how it uses modern tools to achieve its goals. Another, more focused example could be with reference to Christian groups adopting various Jewish practices such as prayer shawls, tefillin, shofars, temple models, and more, and using them to create a particularistic ritualistic cosmology that encourages conformity.
Author Response
Dear Reviewer,
Thank you so much for your very helpful comments on this article.
First of all, I want to apologize for my delay in sending the revised version. The process of writing this article has been quite delayed for a couple of reasons which I will explain briefly here: the paper had already been going through a couple of round of "minor revisions" before receiving "final acceptation.” All was left for me was to take care of language editing, as English is not my first language. Because I don’t have funding for that, the process got delayed, and as I was about to send it, the paper was withdrawn without notice from the system. I was asked to resubmit and told it would just find itself where it was in the process. Instead I received a notice of new reviews asking for “major revisions”. This happened at a time when it was difficult for me to find more time to work on it, and I thought for a moment of dropping it altogether. I am now grateful this happened, because I find that your comments have helped me improve the paper in ways that seem essential and necessary.
At the same time however, I received notice that my step-father was in his last days, and I had to fly back home, where I found myself for a couple of weeks in various parts of France, busy with family matters and a funeral, and far away from libraries or access to some of the very helpful books that were suggested here.
I finally got to get to it, and attached is the revised version, in accordance with your suggestions. I have changed a little bit the introduction to go into the topic more directly. Below are my responses to your comments, point by point (your comments in italics and my responses underneath them), and I have regrouped them thematically for clarity. I go straight to the point for each theme, but every reply should be read with “Thank you for this comment” in the beginning.
Should there be more revisions suggested after this round, I will be able respond more quickly now.
A note about the language: I am sending as is and will have the language edited once I know I have the final version of the text.
Thank you again so much,
MethodologyThere is no methodological chapter (separate and clear, as required by the journal) that explains what was done for the study: what is the method of research? What texts were collected, and how? How was the content analysis carried out? Why were these texts were chosen and not others? How many texts were reviewed?
Thank you, I have included more details about the methodology in the introduction.
- How can one determine that the use of mindfulness meditation among Orthodox Jews is a widespread phenomenon or is gaining momentum? A reference to how often the term appears in forums and Jewish sites could be of help here. It is a test that can be quickly done using technical tools like Google Trends. We must be provided with more information apart from quotes from texts (that it is not clear to us why these were specifically selected).
In the introduction I specify it is an emerging phenomenon (“gaining momentum”) as you suggested. Although I am doing qualitative rather than quantitative research, I have included the various ways that have enabled me to assess objectively this emerging phenomenon (number of google and youtube entries, book publications, institutionalization, empirical observation of the evolution of the field over the past decade etc).
Background of the respondentsWho are the authors of the texts quoted in the article, beyond their religious affiliation and role, something about their religious background that would tell us why they chose to adopt mindfulness meditation for their Jewish religious toolbox?
I have introduced them together a bit more specifically in the introduction, and then have added additional biographic details when I introduce their discourses in the various paragraphs throughout the text.
References suggestedI suggest that the author refer to and expand on Emily Sigalow's work, especially her new book: American JewBu: Jews, Buddhists, and Religious Change. The book describes the process of mutual fertilization that exists in the US between Buddhism and Judaism and addresses a significant portion of the points that are also raised in the article.
I know very well Emily Sigalow’s work indeed, and she knows mine too. She contacted me in 2011 when she was working on her Phd and her book, which indeed describes the process of the integration of Buddhism within contemporary Jewish practice in America via the integration and adaptation of “mindfulness”, develops the pattern I describe my 2015 article which she quotes in her introduction. I was looking forward to her book being published, to this day it is still not available in the libraries where I work but I was able to get hold of it and now that I know it is actually published I will quote it because indeed one cannot speak about these topics without quoting the latest literature on the subject.
How, if anything, does Orthodox writing about mindfulness meditation relate to spiritual practices similar to meditation in Judaism?
I hope to have addressed this question through the categories of “mindfulness through” and “as” Judaism, showing how teachers draw equivalences between the two and consider the first sheds light on the second. I have tried to specify it more in the introduction, and throughout the text I have brought more examples to illustrate that.
Furthermore, what is Jewish meditation? The author can find help on this topic in Tomer Persico's excellent book on Jewish meditation (unfortunately currently only published in Hebrew), in Chava Weissler's writings about the activities of the US Kabala Center, or in ethnographic works by Giller, and by Rubin and Guzmen on prayers of intent as a meditative act among kabbalists.
I was trying to stay focused on the question of use of mindfulness within Jewish practice (the question of cultural import and adaptation), which is a different topic than the question of the Jewish meditation and its reemergence at the forefront of mainstream Jewish life. Of course the two are connected, which is why I did mention Jewish meditation, to reflect on the reason why Jewish teachers and rabbis were using mindfulness, rather than Jewish meditation, to teach Jewish religion. I have however per this request developed a little further the reference to Jewish meditation, and included references to the works abovementioned, while developing with more examples the question of why Jewish teachers feel mindfulness is more helpful to them in understanding and teaching Jewish religion than traditional Jewish meditation.
The article is somewhat weak in theory, although it raises a very intriguing claim that not only does the practice of mindfulness meditation among Orthodox Jews not weaken their faith, but it actually strengthens it and their religious-Jewish affiliation.
Although I have set from the introduction that I was focusing on theory of cultural adaptation in the context of globalization (Hannerz 1990, Turner 2011) , especially through the process of cultural translation (Hall 1992), I have been more specific about the theories I am using here: boundary work (Barth 1969 ) cultural appropriation (Hanh 2008).
Are there any similar instances (at present. The examples presented are historical and there too references to the literature are lacking) where a religious system borrows ideas or tools from outside for its creation?
I have added in the introduction more reference to the literature about historical instances of cultural borrowing in the construction of Jewish culture. And for the contemporary comparative dimension, I am including references to contemporary trends of Muslim and Christian mindfulness.
For example, writing about religious fundamentalism clearly explains how it uses modern tools to achieve its goals. Another, more focused example could be with reference to Christian groups adopting various Jewish practices such as prayer shawls, tefillin, shofars, temple models, and more, and using them to create a particularistic ritualistic cosmology that encourages conformity.
Although I like very much the broadening of the scope that these comparisons open, I’m not sure it would be useful in this article: in the case of fundamentalist groups using modern tools, I am not sure how relevant it is here in the case of Modern Orthodox, because they don’t go against modernity while using its tools just as a strategy. This is what Chabad may be doing, and this is why I have decided to remove the example of Chabad to focus on Modern Orthodoxy- Chabad, your remark made me indirectly realize, is a whole other stance. As for Christian groups using Jewish paraphernalia, although it is always interesting, I am not sure it would bring more to the discussion- I preferred focusing on Christian groups also using mindfulness. If however you feel such comparisons should be added here, I’m happy to hear from references you may have -especially for the second case, and include them.
Reviewer 2 Report
This article is insightful, interesting and original, and is definitely worthy of publication in a journal like Religions. The framing of mindfulness “through” “and” or “as” Judaism is interesting and very helpful.
To improve: The author would do well to include more extensive background on the emergence of Orthodoxy (Katz, Silber, Ferziger) to support the claim that Orthodox rabbis are a kind of gatekeeper – and, even more directly relevant to this thesis, more texture is needed regarding what types of Orthodoxy are being discussed and which cultural milieus they inhabit. The delineation on page three is insufficient. Orthodox rabbis do not currently draw much from Greco-Roman cultures, though – anachronistically – it could be argued that they did so in the past (Orthodoxy is an invention of and a symptom of modernity). What about seeking to identify something quintessentially “American” about these Orthodox practitioners’ approach to religion and spirituality?
James Jacobson Maisels is American but lives in Israel and is very much a part of the Israeli spiritual world.
The author must also read and treat the chapters by Maisels, Slater, and Flam in Arthur Green and Ariel Evan Mayse, A New Hasidism: Branches – which is precisely about this subject. The notion of Neo-Hasidism would be a fruitful one to explore at deeper length.
Some remarks on Jewish meditative traditions (Persico, Reiser, Idel) is needed.
The remark that mahashavah, da’at and sekhel are the opposite of mindfulness (p. 21) is incorrect. Mindfulness includes elements of attunement and consciousness, if not cognitive intellection, which is precisely how da’at is used in many Hasidic works. See Arthur Green’s new essay on that subject.
The claim that all teachers of mindfulness use it as a “spiritual didactic tool… used to better teach Jewish religion” requires nuance. I am sure that some of the Jewish practitioners and teachers of mindfulness see what they do as coming to complement, fix, challenge or upset elements of Judaism.
After all is said and done, can the author give us something bigger to chew on? How does the development and integration of Jewish forms of mindfulness change our understanding of contemporary American Judaism more broadly? Or about
Orthodox appears in both uppercase and lowercase forms. Clarity or consistency is needed. This is true of Mindfulness or mindfulness and Rabbi/rabbi also. Careful copy-editing is needed.
4 l. 132 – word misspelled with ?
Author Response
Dear Reviewer,
Thank you so much for your very helpful comments on this article.
First of all, I want to apologize for my delay in sending the revised version. The process of writing this article has been quite delayed for a couple of reasons, which I will explain briefly here: the paper had already been going through a couple of round of "minor revisions" before receiving "final acceptation.” All was left for me was to take care of language editing, as English is not my first language. Because I don’t have funding for that, the process got delayed, and as I was about to send it, the paper was withdrawn without notice from the system. I was asked to resubmit and told it would just find itself where it was in the process. Instead I received a notice of new reviews asking for “major revisions”. This happened at a time when it was difficult for me to find more time to work on it, and I thought for a moment of dropping it altogether. I am now grateful this happened, because I find that your comments have helped me improve the paper in ways that seem essential and necessary.
At the same time however, I received notice that my step-father was in his last days, and I had to fly back home, where I found myself for a couple of weeks in various parts of France, busy with family matters and a funeral, and far away from libraries or access to some of the very helpful books that were suggested here.
I finally got to get to it, and attached is the revised version, in accordance with your suggestions. I have changed a little bit the introduction to go into the topic more directly. Below are my responses to your comments, point by point (your comments in italics and my responses underneath them), and I have regrouped them thematically for clarity. I go straight to the point for each theme, but every reply should be read with “Thank you for this comment” in the beginning.
Should there be more revisions suggested after this round, I will be able respond more quickly now.
A note about the language: I am sending as is and will have the language edited once I know I have the final version of the text.
Thank you again so much,
The author would do well to include more extensive background on the emergence of Orthodoxy (Katz, Silber, Ferziger) to support the claim that Orthodox rabbis are a kind of gatekeeper –
Yes. I have included these references, plus others.
and, even more directly relevant to this thesis, more texture is needed regarding what types of Orthodoxy are being discussed and which cultural milieus they inhabit.
Indeed. I have specified that it’s about Modern Orthodox Judaism - which, with Heilman and Cohen (1989), I have defined in the introduction.
The delineation on page three is insufficient. Orthodox rabbis do not currently draw much from Greco-Roman cultures, though – anachronistically – it could be argued that they did so in the past (Orthodoxy is an invention of and a symptom of modernity).
They do not currently indeed. I have changed the phrasing in an attempt to be more clear- that rabbis did draw on Greco-roman cultures in the past. I have included in the introduction references to specify that orthodoxy is a modern phenomenon.
What about seeking to identify something quintessentially “American” about these Orthodox practitioners’ approach to religion and spirituality?
I have tried to show this when I was showing the crafting of secular mindfulness by American Jews. I try to be more specific about what is “American” about them.
James Jacobson Maisels is American but lives in Israel and is very much a part of the Israeli spiritual world.
He is indeed; although I would say- from my current experience of working with him in Israel, that if he is structurally part of the Israeli spiritual world, he is also, and perhaps much more, both structurally and culturally, very much part of the Anglo world, both in Israel and in the United States, where he keeps building his presence.
The author must also read and treat the chapters by Maisels, Slater, and Flam in Arthur Green and Ariel Evan Mayse, A New Hasidism: Branches – which is precisely about this subject. The notion of Neo-Hasidism would be a fruitful one to explore at deeper length.
Thank you so much, it’s a recent publication but I got hold of it, and it was very important indeed to include them.
Some remarks on Jewish meditative traditions (Persico, Reiser, Idel) is needed.
I was trying to stay focused on the question of use of mindfulness which different than the question of the Jewish meditation. I did mention Jewish meditation, to reflect on the reason why Jewish teachers and rabbis were using mindfulness, rather than Jewish meditation, to teach Jewish religion. I have however per this request developed a little further the reference to Jewish meditation, and included references to the works abovementioned, while developing with more examples the question of why Jewish teachers feel mindfulness is more helpful to them in understanding and teaching Jewish religion than traditional Jewish meditation.
The remark that mahashavah, da’at and sekhel are the opposite of mindfulness (p. 21) is incorrect. Mindfulness includes elements of attunement and consciousness, if not cognitive intellection, which is precisely how da’at is used in many Hasidic works. See Arthur Green’s new essay on that subject.
I have read and included Art Green’s definition of Da’at, very helpful indeed, thank you. As for mindfulness and cognitive intellection, I did indeed, doing some more research around it, find an article by a Western Buddhist teacher who emphasizes this aspect. Between both, it seems indeed that my argument that they are contradictory falls. I have removed and modified that.
The claim that all teachers of mindfulness use it as a “spiritual didactic tool… used to better teach Jewish religion” requires nuance. I am sure that some of the Jewish practitioners and teachers of mindfulness see what they do as coming to complement, fix, challenge or upset elements of Judaism.
Of course. I am focusing on the didactic tool because this is for a special issue about religious teaching; but of course these teachers use it for themselves, first – and in the ways above-described, as I have tried after this comment to highlight/
After all is said and done, can the author give us something bigger to chew on? How does the development and integration of Jewish forms of mindfulness change our understanding of contemporary American Judaism more broadly? Or about
I have tried to respond to this in the conclusion, in particular by refocusing after reading Green and Mayse on mindfulness as part of the construction of a neo traditional (Orthodox) neo-Hasidic theology ( as opposed to the Jewish renewal neo-Hasidic theology).
Orthodox appears in both uppercase and lowercase forms. Clarity or consistency is needed. This is true of Mindfulness or mindfulness and Rabbi/rabbi also. Careful copy-editing is needed.
Yes, thank you.
Round 2
Reviewer 1 Report
In my humble opinion, the article is ready for publication.
Please Notice that the reference list is messy and does not conform to academic standards.